# Long-Term Survival of Transplanted Autologous Canine Liver Organoids in a *COMMD1*-Deficient Dog Model of Metabolic Liver Disease

**DOI:** 10.3390/cells9020410

**Published:** 2020-02-11

**Authors:** Hedwig S. Kruitwagen, Loes A. Oosterhoff, Monique E. van Wolferen, Chen Chen, Sathidpak Nantasanti Assawarachan, Kerstin Schneeberger, Anne Kummeling, Giora van Straten, Ies C. Akkerdaas, Christel R. Vinke, Frank G. van Steenbeek, Leonie W.L. van Bruggen, Jeannette Wolfswinkel, Guy C.M. Grinwis, Sabine A. Fuchs, Helmuth Gehart, Niels Geijsen, Robert G. Vries, Hans Clevers, Jan Rothuizen, Baukje A. Schotanus, Louis C. Penning, Bart Spee

**Affiliations:** 1Department of Clinical Sciences of Companion Animals, Faculty of Veterinary Medicine, Utrecht University, 3584 CM Utrecht, The Netherlands; L.A.Oosterhoff@uu.nl (L.A.O.); M.E.vanWolferen@uu.nl (M.E.v.W.); c.chen.cn@outlook.com (C.C.); sathidpak@hotmail.com (S.N.A.); k.schneeberger@uu.nl (K.S.); A.Kummeling@uu.nl (A.K.); G.vanStraten@uu.nl (G.v.S.); I.akkerdaas@ziggo.nl (I.C.A.); C.R.Vinke@uu.nl (C.R.V.); F.G.vanSteenbeek@uu.nl (F.G.v.S.); L.W.L.vanBruggen@uu.nl (L.W.L.v.B.); J.Wolfswinkel@uu.nl (J.W.); n.geijsen@hubrecht.eu (N.G.); Rothuize.jan@gmail.com (J.R.); baukje_s@hotmail.com (B.A.S.); l.c.penning@uu.nl (L.C.P.); 2Department of Pathobiology, Faculty of Veterinary Medicine, Utrecht University, 3584 CL Utrecht, The Netherlands; G.C.M.Grinwis@uu.nl; 3Division of Pediatric Gastroenterology, Wilhelmina Children’s Hospital, University Medical Center Utrecht, 3584 EA Utrecht, The Netherlands; S.Fuchs@umcutrecht.nl; 4Hubrecht Institute for Developmental Biology and Stem Cell Research and University Medical Center, Utrecht University, 3584 CT Utrecht, The Netherlands; h.gehart@hubrecht.eu (H.G.); h.clevers@hubrecht.eu (H.C.); 5Hubrecht Organoid Technology (HUB), 3584 CT Utrecht, The Netherlands; r.vries@hub4organoids.nl

**Keywords:** dog, copper, organoids, transplantation, autologous, hepatocytes, cell transplantation, Wilson’s disease

## Abstract

The shortage of liver organ donors is increasing and the need for viable alternatives is urgent. Liver cell (hepatocyte) transplantation may be a less invasive treatment compared with liver transplantation. Unfortunately, hepatocytes cannot be expanded in vitro, and allogenic cell transplantation requires long-term immunosuppression. Organoid-derived adult liver stem cells can be cultured indefinitely to create sufficient cell numbers for transplantation, and they are amenable to gene correction. This study provides preclinical proof of concept of the potential of cell transplantation in a large animal model of inherited copper toxicosis, such as Wilson’s disease, a Mendelian disorder that causes toxic copper accumulation in the liver. Hepatic progenitors from five *COMMD1*-deficient dogs were isolated and cultured using the 3D organoid culture system. After genetic restoration of COMMD1 expression, the organoid-derived hepatocyte-like cells were safely delivered as repeated autologous transplantations via the portal vein. Although engraftment and repopulation percentages were low, the cells survived in the liver for up to two years post-transplantation. The low engraftment was in line with a lack of functional recovery regarding copper excretion. This preclinical study confirms the survival of genetically corrected autologous organoid-derived hepatocyte-like cells in vivo and warrants further optimization of organoid engraftment and functional recovery in a large animal model of human liver disease.

## 1. Introduction

The increasing shortage of donor organs and the morbidity and mortality associated with liver transplantation and subsequent immune suppression underscore the urgent need for novel treatments for patients with liver disease. Liver cell transplantation can be a less invasive alternative to liver transplantation, especially in cases of metabolic liver diseases. Partial liver repopulation with healthy hepatocytes has been successful for human Crigler–Najjar syndrome, phenylketonuria and urea cycle defects [1,2,3,4]. The transplanted hepatocytes, however, were obtained from a donor liver and recipients required lifelong immunosuppression. More importantly, primary hepatocytes do not proliferate in vitro and rapidly lose essential hepatocyte functions in culture [5,6]. Similarly, the effect of hepatocyte transplantations is lost after months, putatively due to the limited lifespan of hepatocytes [7].

In contrast, adult stem cells of the liver cultured as organoids are highly proliferative and display a progenitor phenotype during in vitro expansion [8]. In addition, these liver organoids can be differentiated towards hepatocyte-like cells, and several groups have reported the engraftment of mouse, rat and human liver organoid-derived cells into rodent liver disease models [8,9,10]. This has prompted the question of whether human liver organoids could be a therapeutically relevant cell source for hepatocyte transplantation in patients with metabolic liver disease [11]. In most metabolic liver diseases, an estimated 2–5% repopulation of the liver with normal non-proliferating hepatocytes is sufficient for clinical recovery [12]. Thus, only several rounds of proliferation of transplanted cells in vivo would be required to reach clinically relevant repopulation levels. But before human liver organoids can be applied in a first-in-man transplantation study, the translational aspects should be verified in a large animal model [13,14]. This specifically allows evaluation of (1) the optimal route of cell administration (i.e., intraportal infusion, a clinically applied and safe route in man, is not feasible in mouse and rat), (2) autologous transplantation with biopsy-derived patient-specific stem cells that are genetically corrected, and (3) longitudinal follow up in the same individual for cell tracking and safety evaluation. Furthermore, clinical efficacy should ideally be investigated in cases of spontaneous liver disease that resemble the human clinical situation and physiology. 

Dogs have naturally occurring liver diseases and mechanisms of canine liver disease and regeneration show striking similarities with humans on both a molecular and cellular level [15,16,17]. Canine copper storage disease is caused by a deletion of exon 2 of the copper metabolism domain containing 1 (*COMMD1*) gene that results in impaired copper excretion from hepatocytes into the bile [18,19,20]. *COMMD1*-deficient dogs develop hepatic copper storage disease and chronic hepatitis similar to human Wilson’s disease [19,21,22]. Therefore, canine COMMD1-linked copper toxicosis is a valuable preclinical disease model to study functional recovery by means of liver organoid transplantations. Our group has established and extensively characterized a canine liver organoid culture system and demonstrated that genetically corrected *COMMD1*-deficient organoids *in vitro* display restored copper excretion [23]. In this study, we transplanted cells from autologous gene-corrected canine liver organoids in *COMMD1*-deficient dogs and used a routing of transplantation that can be easily extrapolated to human clinical application. We evaluated engraftment, repopulation and functional recovery of liver disease and report on the long-term (max 2 years) survival of these cells in a relevant canine model for inherited copper toxicosis, such as Wilson’s disease.

## 2. Materials and Methods

### 2.1. Study Design

Patient-specific canine *COMMD1*-deficient autologous liver stem cells were extracted from five dogs and the genetic defect was corrected using a lentivirus (4 dogs corrected and 1 vehicle control). Organoids were then transplanted back into the liver, via the portal vein or intrahepatic injection, of the respective canine patients (Figure 1). Transplantation effects were measured relative to vehicle control (Appendix A). This study adheres to the Animal Research: Reporting of In Vivo Experiments (ARRIVE) guidelines.

### 2.2. COMMD1-Deficient Dogs

All studies were approved by the Utrecht University ethical committee, as required under Dutch legislation (DEC2014.III.04.039, DEC2014.III.12.112). Five *COMMD1*-deficient Beagle–Bedlington terrier crossbreed dogs (details in Appendix A) were used from a breeding colony harboring a deletion in exon 2 of the *COMMD1* gene and diagnosed with chronic hepatitis due to copper storage disease [21]. Normal canine liver samples for pilot experiments were obtained from fresh canine cadavers used in non-liver related research (surplus material, Utrecht University 3R-policy). 

### 2.3. Biliary Duct Isolation, Autologous Liver Organoid Culture, Lentiviral Transduction and Harvest

Three months before transplantation, a biliary ^64^Cu excretion study was performed and liver biopsies were taken to obtain autologous liver stem cells residing in biliary duct fragments. Canine liver organoid culture and lentiviral transduction was performed as described before [23]. Briefly, two 14G Tru-cut liver biopsies were minced and digested in DMEM with 1% *v*/*v* FCS containing 0.3 mg/mL collagenase type II and 0.3 mg/mL dispase (all from LifeTechnologies, Carlsbad, CA, USA) at 37 °C. Biliary duct fragments appeared in the supernatant after two to four hours. Ducts were plated in Matrigel (BD Biosciences, Erembodegem, Belgium) and expansion medium was added to the wells after gelation. Organoids were passaged by mechanical disruption once a week at a 1:6 split ratio. 

At passage two, organoids were enzymatically dissociated and lentiviral (LV) transduction with a pHAGE2-EF1a-COMMD1-DsRed-PuroR or a pHAGE2-EF1a-COMMD1-eGFP-PuroR construct was performed using spinoculation as described earlier [23]. Culture was continued with puromycin to select for transduced cells. Autologous gene-corrected liver organoids were expanded for transplantation in 12 well plates (Greiner, Alphen aan den Rijn, The Netherlands) in 100 µL Matrigel droplets per well and a total of 324 wells were cultured for each dog.

To induce differentiation towards hepatocyte-like cells, 25 ng/mL BMP7 (Peprotech, London, United Kingdom) was added to the expansion medium after the last passage. Four days after the last passage, Wnt-conditioned medium, ROCK inhibitor and Noggin were withdrawn from the medium and BMP7 treatment was continued. Six days after the last passage, nicotinamide, R-spondin-1-conditioned medium and FGF10 were also withdrawn from the medium, BMP7 was continued and 100 ng/mL FGF19 (R&D Systems, Abingdon, United Kingdom), 10 µM DAPT (Selleckchem, Huissen, The Netherlands) and 30 µM dexamethasone (Sigma-Aldrich, Zwijndrecht, The Netherlands) were added (differentiation medium, DM). Culture in DM was continued for eight to nine days. Differentiation of DM (DM, n = 2 dogs) conditions before transplantation was confirmed by gene expression profiling indicating a decrease in stemness marker (LGR5) and an increase in hepatic markers (HNF4A and ALB) after differentiation, see Appendix A.

On each consecutive transplantation day (day 0, day 1, day 2), approximately 108 wells of undifferentiated (EM, n = 2 dogs) or differentiated (DM, n = 2 dogs) autologous pHAGE2-EF1a-COMMD1-DsRed-PuroR-transduced liver organoids were harvested just prior to transplantation. 

### 2.4. Microbead Perfusion of Canine Liver 

A pilot experiment was performed to determine minimum cell size for intraportal delivery of cells in a canine liver. A heparinized cadaveric canine liver was infused with 10 µm red fluorescent microbeads (Life Technologies) in HBSS (Life Technologies). Infusion was given via the portal vein using an inflated balloon catheter (MILA, Utrecht, The Netherlands) to prevent backflow. The inferior vena cava was ligated caudal to the liver and cannulated cranial to the liver to collect all flow through. Infusion with HBSS was continued for an additional 15 min after microbead infusion. Flow through was centrifuged at 250× *g* for 5 min. Liver was sampled using wedge biopsies and Tru-cut biopsies. Fresh 1 mm thick slices were cut from the wedge biopsies for direct evaluation of native fluorescence using an Olympus IMT-2 microscope (Leiderdorp, The Netherlands). Tru-cut biopsies were frozen in TissueTek (Sakura, Alphen aan den Rijn, The Netherlands), cryosections were prepared and immediately microscopically evaluated for the presence of microbeads. 

### 2.5. Partial Hepatectomy and Portal Catheter Implantation

On the first day of transplantation (day 0), dogs were anesthetized for a partial hepatectomy and placement of a vascular access system in the portal vein. Using a midline celiotomy approach, a left lateral hepatic lobectomy was performed, resulting in approximately 20% reduction in liver mass. A permanent Port-A-Cath (PAC, Smiths Medical, Rosmalen, The Netherlands) system was implanted into the portal vein to provide non-invasive access for repeated intraportal delivery of cells [24,25]. The catheter was inserted in either a jejunal or splenic vein; the tip was advanced into the portal vein and placed 1–2 cm caudally to the liver hilum. A gripper needle was placed percutaneously into the portal and was removed five days after surgery. 

### 2.6. Transplantation of Organoid-Derived Liver Cells by Intrahepatic Injection

Three dogs were transplanted by means of intrahepatic injections (Appendix A). In two dogs, intrahepatic transplantation was performed in the same surgical procedure as the intraportal transplantation (day 1). One dog was retransplanted with intrahepatic injections two years after intraportal transplantation during a second celiotomy procedure. For intrahepatic transplantation, pHAGE2-EF1a-COMMD1-eGFP-PuroR transduced autologous liver organoids were cultured under undifferentiated (EM) and differentiated (DM) conditions. Organoids were isolated from Matrigel using cold advanced DMEM/F12 (LifeTechnologies) and mechanically dissociated into small fragments (Appendix A) or enzymatically digested to single cell level with TrypLE select enzyme (LifeTechnologies). Organoid-derived liver cells were resuspended in 0.9% *w*/*v* NaCl with 10% *v*/*v* autologous serum and transferred to serum-precoated Eppendorf tubes. Immediately before injection, fragments were transferred to a serum-precoated syringe with a 12 mm 29G needle (Kruuse). Injections were made into the liver during a celiotomy procedure. Per injection site (n = 2 technical replicates per condition), 5–9 injections of 50 µL each spaced 2 mm apart were administered slowly into the liver at a depth of 10–12 mm. Injection sites were marked with electrocautery and polypropylene sutures (Appendix A). Intrahepatic vehicle injections served as negative control and post-mortem established organoid-derived liver cell injections served as positive control. One week (n = 2 dogs) and one month (n = 1 dog) after intrahepatic transplantation dogs were euthanized and the liver was harvested. All injection sites were sampled by resecting a 1 × 1 cm piece of liver between the polypropylene sutures, cutting 2 cm deep into the parenchyma. The 1 × 1 × 2 cm rectangular liver specimen was then cut into four pieces to create section levels at 0.5, 1, 1.5 and 2 cm liver depth. Liver tissue was fixed in 10% neutral buffered formalin for 24 h, transferred to 70% ethanol and embedded in paraffin.

### 2.7. Transplantation of Organoid-Derived Liver Cells by Intraportal Delivery 

Cells were transplanted on three consecutive days via the portal vein (n = 4 dogs, 1 vehicle control dog) and via intrahepatic injections (n = 3 dogs, also transplanted via portal vein). To provide a regenerative stimulus, a partial hepatectomy was performed on the first day (day 0) of transplantation [26], and the first fraction of organoid-derived liver cells were transplanted intraoperatively to enable visual monitoring of infusion via the PAC and possible effects on abdominal organs, in particular, for portal hypertension. On days 1 and 2, the second and third fractions of organoid-derived liver cells were transplanted without sedation under abdominal Doppler ultrasound guidance (Philips HD11). 

### 2.8. Immune Suppression

Dogs were treated with cyclosporine (ASTfarma, Oudewater, The Netherlands) to prevent potential rejection of genetically modified autologous cells. Dosage was readjusted based on weekly cyclosporine plasma levels as measured 2 h after oral administration (peak plasma concentration). Treatment was initiated one day before transplantation and continued for 1 month at 6.25 mg/kg q12h (0.6–1.0 mg/L peak plasma concentration). Dosage was then lowered to 3.13 mg/kg q12h (0.3–0.6 mg/L peak plasma concentration) and treatment continued for an additional two months. As gastrointestinal side effects (anorexia, vomiting, diarrhea) were observed at high dose oral cyclosporine, in two dogs cyclosporine treatment was initiated three weeks before transplantation with an increasing dosage regimen. Side effects were treated with antiemetics (metoclopramide, ondansetron, maropitant) and an antacid (omeprazole). 

### 2.9. Follow-Up Measurements: Liver Biopsies, Blood Analysis, Staining and Biliary ^64^Cu Excretion Measurements

Follow-up measurements consisting of blood analysis, liver biopsies and biliary ^64^Cu excretion studies were performed one week (n = 5), one month (n = 4), three months (n = 3), six months (n = 3), nine months (n = 3), one year (n = 3) and two years (n = 1) after transplantation. One dog (DM2) was re-transplanted (with COMMD1-eGFP lentiviral construct) two years after the beginning of the study by means of intrahepatic injections. Seven days after intrahepatic transplantation, the dog was euthanized, and the entire liver was harvested. For post-transplantation cell tracking, the liver was biopsied with a Tru-cut automatic biopsy instrument (Angiotech, Stenloese, Denmark) in sedated dogs (methadone 0.5 mg/kg IM, propofol 1–4 mg/kg IV, lidocaine local abdominal block). At each time point (Appendix A), four biopsies were taken and when possible, two each from two separate lobes. Biopsies were fixed in 10% neutral buffered formalin for 4 h, transferred to 70% ethanol and embedded in paraffin. At each biopsy moment, a small volume of blood was withdrawn and plasma activity levels for liver enzymes (ALP, ALT) were determined. Excretion of exogenously administered ^64^Cu into the bile was investigated. Dogs were kept in a metabolic cage for the short duration of ^64^Cu excretion study. A 1.5 mg copper wire was irradiated for 10 h in a reactor, providing an activity of approximately 45 MBq/mg. Copper was dissolved in 50 µL concentrated HNO_3_ (10.3M) and neutralized with 1.3 mL NaOH (0.5M). A dose of 0.003 mg/kg copper was prepared in 2.5 mL of autologous heparinized plasma, corresponding to 10% of the copper plasma pool. Dogs received methadone (0.3 mg/kg IM, repeated dose after three hours) to close the sphincter of Oddi and ^64^Cu was administered into the cephalic vein. After six hours, dogs were sedated with methadone (0.5 mg/kg IM) and propofol (1–4 mg/kg IV). Under ultrasound guidance, the gallbladder was punctured and emptied by aspiration. Activity of ^64^Cu in the bile was measured with a gamma counter and corrected for decay between administration of the IV dose and measurement of the bile. Routine hematoxylin and eosin (H & E) staining was performed and liver sections were histologically analyzed for hepatitis grade (0–5) and fibrosis stage (0–4) [27] by a board-certified veterinary pathologist. For immunostainings sections of paraffin-embedded liver samples and organoids were dewaxed and rehydrated using a graded ethanol series. Immunocytochemistry and histochemistry staining (ICC/IHC) for COMMD1, eGFP and DsRed were performed as described in Appendix A. Normal dog liver was used as positive control for COMMD1 immunohistochemistry. Dog liver injected post-mortem with eGFP- or DsRed-transduced organoids were used as positive control for eGFP and DsRed immunohistochemistry, respectively. Elastica van Gieson staining were stained with Lawson’s solution combined with Mayer´s Haematoxylin and van Gieson at the department of the Faculty of Veterinary Medicine for visualization of collagen and elastic fibers. Double immunohistochemistry was performed for COMMD1-Ki67 and DsRed-MRP2 using a serial staining. After antigen retrieval, endogenous peroxidase and phosphatase activities were blocked (DAKO) and sections were incubated with 20% normal horse serum (Vector). The mouse primary antibody was incubated for 1 h at room temperature (RT). Endogenous biotin was blocked (Genemed Biotechnologies) and sections were incubated with horse anti-mouse biotinylated IgG 1:200 (Vector) for 45 min at RT. Sections were washed and subsequently incubated with 10% normal goat serum (Sigma-Aldrich). The rabbit primary antibody was incubated overnight at 4 °C. Sections were incubated with streptavidin alkaline phosphatase 1:2000 (Vector) for 45 min at RT and Vector Red (Vector, red chromogen) was incubated for 20 min at RT. Sections were washed and then incubated with HRP-labeled goat anti-rabbit antibody (Envision, DAKO) for 45 min at RT. Sections were washed and incubated with 3,3′-diaminobenzidine (DAKO, brown chromogen) for 5 min at RT. Sections were counterstained with hematoxylin and mounted in Vectamount (Vector). As a technical negative control, the primary antibody was omitted. Sections were analyzed and imaged with an Olympus microscope (CKX41) combined with a Leica DFC425C camera.

### 2.10. Repopulation Counts 

Repopulation with DsRed-positive cells after transplantation was determined in liver samples obtained by necropsy (dog EM1: 1 week; dog EM2: 1 month; dog DM2: 2 years) or biopsy (dog DM1: 1 year). We counted 700–1100 separate fields in necropsy samples and 33 fields in biopsies at a 200× magnification. The number of DsRed-positive cells was related to an average total hepatocyte count per field to calculate liver cell repopulation percentage. For all counted cells, histological distribution was scored as either periportal, parenchymal or pericentral and engraftment in either fibrous or non-fibrous tissue was scored. 

## 3. Results

### 3.1. Autologous Gene-Corrected Liver Organoids Can Be Sufficiently Expanded for Transplantation within 12 Weeks

Patient-specific canine *COMMD1*-deficient organoids were successfully cultured from liver biopsies (Figure 1A, Appendix A). After lentiviral (LV) transduction with a pHAGE2-EF1a-COMMD1-DsRed-PuroR or a pHAGE2-EF1a-COMMD1-eGFP-PuroR construct, followed by puromycin selection, all organoids in culture acquired a red (DsRed) or green (eGFP) fluorescent phenotype, respectively (Appendix A). We confirmed the presence of the COMMD1 protein in liver organoids after transduction by immunocytochemistry (Appendix A) and induced organoid differentiation after expansion by changing medium composition from expansion medium (EM) to differentiation medium (DM) [23]. After the initial biopsy, a total cell dose of 4.4–9.3 × 10^8^ with cell viability ranging from 94% to 99% was reached within 12 weeks of culture, which was sufficient for transplantation (Figure 1B). Based on an average liver mass of 350 g and the hepatocyte density of a canine liver [28], this number of cells would constitute 0.6–1.2% repopulation. 

### 3.2. Organoid Fragments and Single Cells Survive only Short Term (7 days) after Intrahepatic Injection, Irrespective of Differentiation Status; Organoid Fragments but not Single Cells Induce De Novo Stroma Formation

To investigate whether organoid differentiation status and organoid size (single cells vs. fragments) would affect survival of transplanted cells in the liver, we injected autologous COMMD1-eGFP-transduced EM and DM organoid-derived single cells or fragments directly into the liver (n = three dogs) at multiple injection sites. Injection sites were harvested one week and one month after transplantation. We observed eGFP-positive cells, fragments and organoid-like structures in sections of the liver at one week, but not one month after transplantation (Figure 2). Injected EM and DM fragments, but not single cells, were consistently embedded in newly formed fibrous tissue. Elastica van Gieson staining indicated a sometimes circular deposition of loosely arranged, non-birefringent collagen fibrils around the organoid fragments in the liver; elastin fibers were absent. In vehicle-injected livers and post-mortem organoid-injected control liver samples, we did not see stromal deposition around injected fragments (Appendix A). Because an in vivo profibrogenic phenotype of transplanted organoid fragments is highly undesirable, we decided to continue the transplantations with organoid-derived single cells, and to improve the survival beyond one week by choosing an alternative route of cell delivery.

### 3.3. Organoid-Derived Cells Can Be Safely and Repeatedly Infused via the Portal Vein

Previous human hepatocyte and rodent organoid transplantations were performed via the spleen in order to provide significant cell numbers for repopulation [1,2,3,8,9,10,29]. To investigate the feasibility of organoid-derived liver cell transplantation via the portal vein in dogs, we conducted a pilot experiment to check whether the cells remain in the liver upon portal delivery. In canine hepatic scintigraphy, ^99^Tc-labeled macro-aggregated albumin particles of 10–90 µm in size are routinely used to quantify portal blood flow since they lodge in the liver sinusoids [30]. Single cells from organoids have a size of approximately 10–12 µm (Appendix A). Thus, to verify whether this cell size is large enough to prevent systemic flow through, we perfused a fresh cadaveric canine liver with 10 µm fluorescent microbeads and found that over 99.9% of the microbeads got trapped in the liver upon portal delivery. In addition, we observed that the microbeads were arranged as branching strings in liver sections, indicative of a position lodged in the intrahepatic vasculature (Appendix A). 

Subsequently, autologous COMMD1-DsRed-transduced organoid-derived liver cells from both undifferentiated (dogs EM1 and EM2) and differentiated (dogs DM1 and DM2) cultures were transplanted through the portal vein as single cells via a permanent catheter with a subcutaneous port (Figure 3A). One dog received vehicle control (dog veh ctr). Providing a regenerative stimulus to the liver at the time of transplantation is a prerequisite for donor cell proliferation in rodent hepatocyte transplantations [26,31]. We therefore performed a liver lobectomy on the first day of the 3 day transplantation procedure (Figure 3B) and detected Ki67-positive hepatocytes in the biopsies after transplantation, indicative of a local regenerative environment. For transplantation, the total cell dose was divided into three fractions and given on three consecutive days to enhance engraftment. Portal infusion of either vehicle or cells was not associated with changes in heart rate or mean arterial blood pressure (data not shown) and could be performed without sedation on days two and three under Doppler ultrasound guidance (Figure 3C). Plasma concentrations of the liver enzymes alkaline phosphatase (ALP) and alanine aminotransferase (ALT) were within reference range or minimally elevated before transplantation and increased, as expected, in all dogs on the days after liver lobectomy (Figure 3D). Values decreased to pre-transplantation levels within one month and remained stable during the follow-up period. Portal pressure increased after cell infusion to a maximum of 189% of pre-transplantation level (Figure 3E); vehicle control infusion did not result in increased portal pressure. These data suggest that repeated transplantation of organoid-derived single cells can be performed safely through the portal vein. 

### 3.4. Gene-Corrected Organoid-Derived Cells Engraft and Survive in the Liver for Up To Two Years after Intraportal Delivery

In order to assess engraftment, we obtained liver samples by either biopsy or necropsy and evaluated them for the presence of transplanted cells for up to two years post-transplantation. We observed DsRed-positive cells in liver sections of all transplanted dogs (EM1: 1 wk; EM2: 1 wk, 1 mo; DM1: 1 mo, 6 mo, 1 yr; DM2: 1 mo, 3 mo, 9 mo, 2 yr) (Figure 4A). The transplanted cells were mainly identified as single cells and not as cell clusters and tumor formation was not detected. These data indicate that gene-corrected organoid-derived cells can survive in vivo for up to two years post-transplantation. Repopulation was low and ranged between 0.015% and 0.13% and was independent of differentiation stage (Figure 4B). Histological detection of transplanted cells showed that they were mainly distributed in parenchymal and pericentral areas but not in periportal areas (Figure 4C). Stage of hepatic fibrosis varied between dogs but was stable throughout the study [27]. Interestingly, in dogs EM2 and DM2, which exhibited the highest stage of fibrosis (stage 2–3 both pre- and post-transplantation), the majority of the engrafted cells localized within the fibrous tissue. This fibrous tissue was consistent with pre-existing stroma, suggesting preferred cellular engraftment within fibrous tissue or altered local hemodynamics that influence cellular entrapment. These data suggest that engraftment may be influenced by local in vivo factors, including fibrous tissue.

### 3.5. Transplanted Organoid-Derived Cells Do Not Show Full Maturation and Functional Integration In Vivo

To investigate whether the transplanted cells proliferated and/or differentiated after engraftment in vivo, we performed double staining for COMMD1-Ki67 and DsRed-MRP2 on liver sections post-transplantation. Transplanted cells were only sporadically Ki67 positive in liver samples obtained 1 wk post-transplantation (only EM cells), but not at later time points (Figure 5A). Transplanted cells did not stain positive for hepatocyte canalicular marker MRP2 in any of the samples, suggesting insufficient differentiation and/or integration within the hepatocyte cords (Figure 5B). To evaluate whether transplanted *COMMD1*-corrected cells made any functional contribution to copper metabolism, biliary excretion of exogenously administered ^64^Cu was measured before and after transplantation. At the start of the study, biliary ^64^Cu after six hours was <1% of the total injected dose in all dogs, which is consistent with copper storage disease [32]; despite the successful engraftment, the repopulation level was not yet sufficient to significantly impact biliary copper excretion (Figure 5C). These data suggest that while organoid-derived cells engraft, further proliferation and functional integration are, for some reason, inhibited. 

## 4. Discussion

Liver cell transplantation may provide a good and less invasive alternative to whole-organ transplantation. For successful clinical application, sufficient numbers of (gene-corrected autologous) cells need to be established and tested for clinical effects and safety using the clinically intended transplantation method in a large animal model with a disease that closely parallels the human condition. In this study, we demonstrated use of autologous gene-corrected liver organoids for cell transplantation in a canine *COMMD1*-deficient model of copper storage disease, which closely resembles human Wilson’s disease.

Derivation from the already insufficient number of donor livers and inability to expand primary hepatocytes in vitro, makes this cell type poorly suitable for therapeutic application [33]. In contrast, organoids can be readily expanded [10,11,23] and are easy to maintain in culture, enabling genetic correction. As such, transplantation of autologous gene-corrected organoid-derived cells can avoid immune rejection and eliminate the need for long-term immunosuppression. 

An important consideration for transplantation is organoid size. Upon intrahepatic injection, we did not detect differences in cell survival between organoid-derived single cells or organoid fragments. Interestingly, we did observe fibrous tissue formation around organoid fragments, but not around organoid-derived single cells. We postulate that this may be due to local tissue reaction. Alternatively, the organoids themselves produce extracellular matrix components, as has been previously described [34,35]. Since inducing fibrosis would be clinically undesirable, we chose to continue with organoid derived single cell preparations. 

For human and canine hepatocyte transplantations, administration through the portal vein is an elegant, relatively non-invasive, safe and already clinically applied procedure that allows rapid dispersion over the liver [24,25,29]. Experimental mouse, rat and human liver organoid transplantations in rodents have been performed by intrasplenic injection, which also result in cellular engraftment in the liver via the splenic vein which feeds the portal vein [9,10]. Intraportally transplanted cells are known to embolize in the sinusoids, migrate through the fenestrae in the endothelium and integrate into the hepatic parenchyma. Cells remaining in the vascular lumen are cleared by phagocytosis within 24–48 h [36,37,38]. In human hepatocyte transplantation, engraftment efficiency is low, often requiring repeated infusions of cells [5,38]. We tested administration through the portal vein and divided the total transplanted cell dose over three infusions on consecutive days in an effort to promote hepatic engraftment and were able to infuse high numbers of organoid-derived single cells directly into the portal bloodstream. Our results demonstrate that we can repeatedly deliver organoid-derived cells via the portal vein in an efficient and non-invasive manner. Moreover, portal hypertension is a known potential complication in canine and human hepatocyte transplantations and precludes infusion of high cell numbers [29,38]. Our procedure was safe (see also Appendix A) and was performed without sedation and resulted in only minor and transient elevation in portal blood pressure. Repetitive organoid derived cell transplantations through the portal vein may thus be easily translated into a human clinical therapy. 

Using this procedure, we observed engraftment and survival in the liver of gene-corrected cells for up to two years after transplantation. However, the repopulation level was low and suggests poor engraftment efficiency. Based on a calculated 0.6–1.2% transplanted cells of all cells in the canine liver and the observed 0.015–0.13% liver repopulation, engraftment efficiency most likely ranged from 1% to 10% of all injected cells. This is consistent with observations in both mouse and rat organoid transplantations, where low engraftment and liver repopulation efficiencies have been reported. For mouse, 5 out of 15 successfully transplanted mice resulted in 0.1–1% repopulation [9]. Rat organoid transplantation was successful in 3 out of 7 transplanted animals [10]. Conceivably, unsuccessful transplantation in a rodent could be due to the result of the intrasplenic injection route which requires the cells to leave the splenic pulp to reach the portal bloodstream [39]. Future studies will need to be conducted to evaluate whether increasing the number of infusions will result in a higher cumulative repopulation percentage. 

Repopulation of the liver with transplanted cells to clinically relevant levels requires in vivo proliferation. We provided a regenerative stimulus in the form of a liver lobectomy and detected a local regenerative environment; however, further analyses of transplanted liver sections showed that cells were not configured as colonies or clusters and were only at early time points occasionally positive for Ki67. When we stained for the hepatocyte marker MRP2, the transplanted cells were negative, suggesting a progenitor- rather than a hepatocyte-like phenotype. Because progenitor cells do not generally contribute to hepatectomy-induced regeneration [14,40], this may explain the lack of proliferative response from the transplanted organoid-derived cells. Moreover, we did not observe clinical improvement of copper metabolism. Our findings indicate that although transplanted organoid-derived cells engraft and survive, they do not proliferate or seem to differentiate to functional hepatocytes in vivo. If left untreated, COMMD1 deficiency-related copper toxicosis will result in cirrhosis in 42 months [21]. With the current data, no survival benefit could be shown.

When human hepatocytes are transplanted, the clinical effect is generally short lived and little is known about the in vivo behavior of these cells. Although it has been shown that donor cells persist long-term in liver biopsies one year after transplantation [3], similar to our results, evidence of in vivo proliferation is lacking. This has precluded clinical application of human hepatocyte transplantation for permanent recovery of metabolic liver disease and currently limits clinical use to bridging a patient to liver transplantation [6,38]. The higher engraftment of hepatocytes compared to organoid cells, and the short-term functional recovery after hepatocyte transplantation may be explained by their advanced hepatocyte maturation state. Differentiation of liver organoids and iPS cell-derived liver cells in vitro are suboptimal in several species [8,9,10,41,42], and thus it is essential to optimize this prior to transplantation. Future research should focus on liver organoid differentiation to improve engraftment, repopulation and short- and long-term functional recovery from liver disease.

The number of transplanted dogs in our study was low. This is an obvious limitation to such large and therefore costly experiments compared to rodent studies. Moreover, ethical issues, especially once dogs are included, limit large numbers of experimental animals. However, the large size of the canine liver allowed us to evaluate both intrahepatic and intraportal transplantations in the same procedure and permitted for a longitudinal study with pre-transplantation control measurements for each individual dog. These are important advantages over rodent models, in which repeated liver biopsies for autologous organoid culture, cell tracking and safety evaluation are not feasible. Moreover, our dogs represent a highly relevant animal model, because essential molecular pathways and cells that contribute to regeneration are similar in dogs and humans with naturally occurring liver disease [15,16,17]. Hence, preclinical studies with autologous canine liver organoid transplantation will likely have a high predictive value for the safety and efficacy of autologous human liver organoid transplantations. 

In conclusion, we demonstrate that intraportal transplantation of high organoid-derived cell numbers is safe in a canine model of copper associated liver disease. Upon gene correction and subsequent autologous cell transplantation, cells survive long-term in the liver. Importantly, canine patients can sustain this minimally invasive procedure during repeated infusions, which is currently used for human hepatocyte transplantations. Because canine liver disease closely mimics the human condition, this model will enable future studies to improve functional repopulation of the liver. 

## Figures and Tables

**Figure 1 cells-09-00410-f001:**
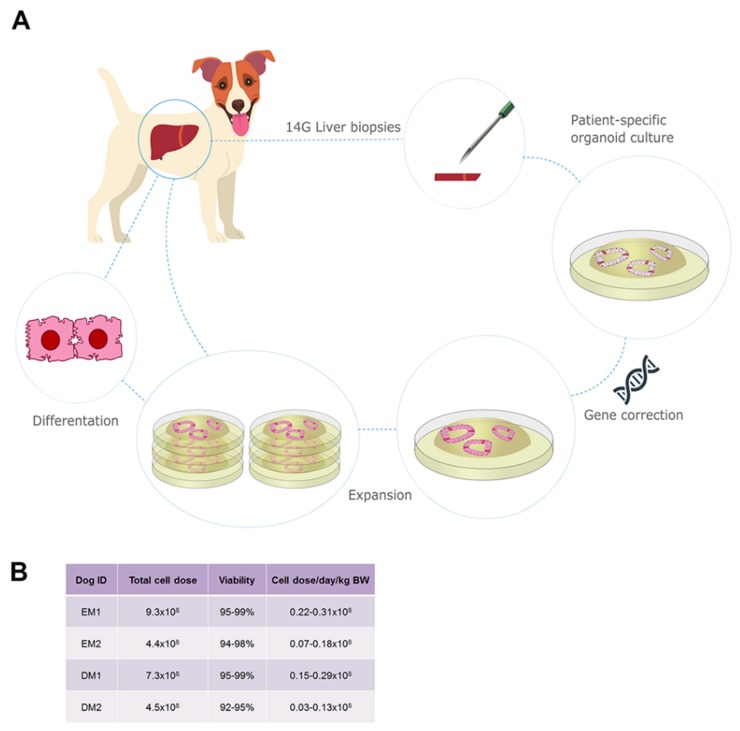
Establishment, genetic correction and expansion of autologous canine liver organoid culture for transplantation. (**A**) Schematic representation of organoid culture for autologous transplantation. From four *COMMD1*-deficient dogs, liver biopsies were taken for isolation of biliary duct fragments and patient-specific organoid culture. Organoids were genetically corrected to incorporate the full length canine COMMD1 cDNA and expanded in culture. Transplantation of autologous organoids was performed with cells in an undifferentiated state and after differentiation towards hepatocyte-like cells. (**B**) A total cell dose of 4.4–9.3 × 10^8^ cells was reached within 12 weeks of culture, which was sufficient for transplantation.

**Figure 2 cells-09-00410-f002:**
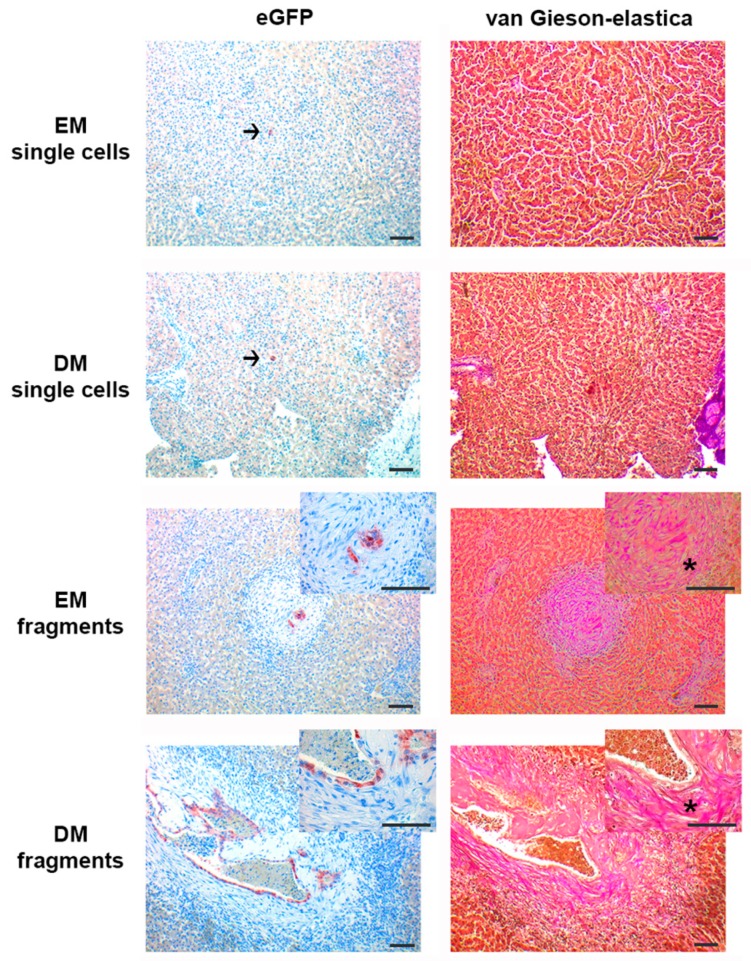
Organoid engraftment after intrahepatic injection. Representative images of serial immunohistochemistry stainings for eGFP (red chromogen) and Elastica van Gieson staining of liver one week after injection with undifferentiated (EM) and differentiated (DM) organoid-derived single cells vs. organoid fragments. All conditions showed engraftment (arrows highlight engrafted single cells). Extracellular matrix deposition (*) can be seen surrounding EM and DM organoid fragments, but not around single cells. Scale bars represent 50 µm.

**Figure 3 cells-09-00410-f003:**
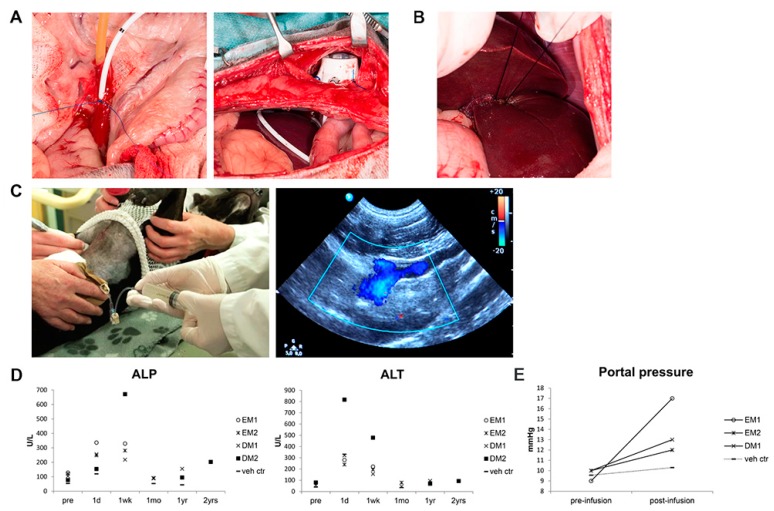
Repeated transplantation of organoid-derived cells via the portal vein. Organoids were dissociated to single cells and transplanted by intraportal delivery into the liver on three consecutive days. On the first day (**A**), a permanent catheter was placed in the portal vein and connected to a subcutaneous port and (**B**) a left lateral hepatic lobectomy was performed. (**C**) On day two and three, cells were transplanted non-invasively via the catheter in unsedated dogs under Doppler ultrasound guidance. (**D**) Plasma activity levels of liver enzymes alkaline phosphatase (ALP, ref. <89 U/L) and alanine aminotransferase (ALT, ref. <70 U/L) increased after hepatic lobectomy but returned to pretreatment levels within one month. pre: pre-transplantation; d1: one day; 1 wk: one week; 1 mo: one month; 1 yr: one year; 2 yrs: two years post-transplantation. (**E**) Portal pressure increased in all dogs after infusion of cells, but not after vehicle infusion (vehicle control dog).

**Figure 4 cells-09-00410-f004:**
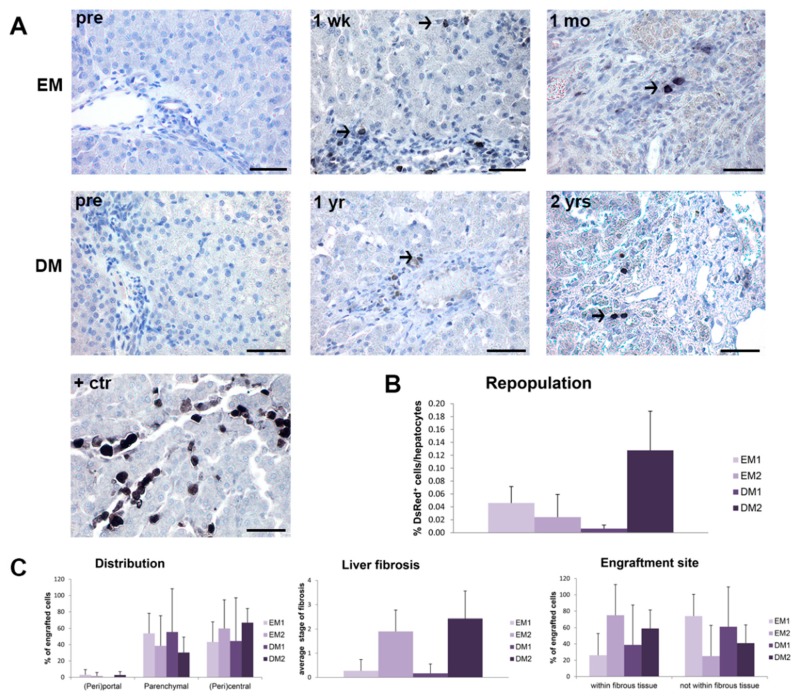
Engraftment and long-term survival of organoid-derived liver cells after intraportal delivery. Four *COMMD1*-deficient dogs were transplanted via the portal vein and liver was sampled at various time points after transplantation for cell tracking purposes. (**A**) Representative images of immunohistochemical staining for DsRed in liver sections of undifferentiated (EM) or differentiated (DM) organoid-derived liver cells pre-transplantation (pre) and one week (1 wk: dog EM1), one month (1 mo: dog EM2), one year (1 yr: dog DM1) and two years (2 yrs: dog DM2) post-transplantation. Normal dog liver injected post-mortem with DsRed-transduced organoid-derived liver cells was used as positive control (+ctr). (**B**) Repopulation of liver with DsRed-positive cells expressed as percentage of total hepatocyte count. (**C**) Histologic distribution of transplanted cells and engraftment in either fibrous or non-fibrous tissue was determined and expressed as percentage of engrafted cells. Average stage of fibrosis was scored in liver sections after transplantation (2–10 sections per dog). Scale bars represent 50 µm.

**Figure 5 cells-09-00410-f005:**
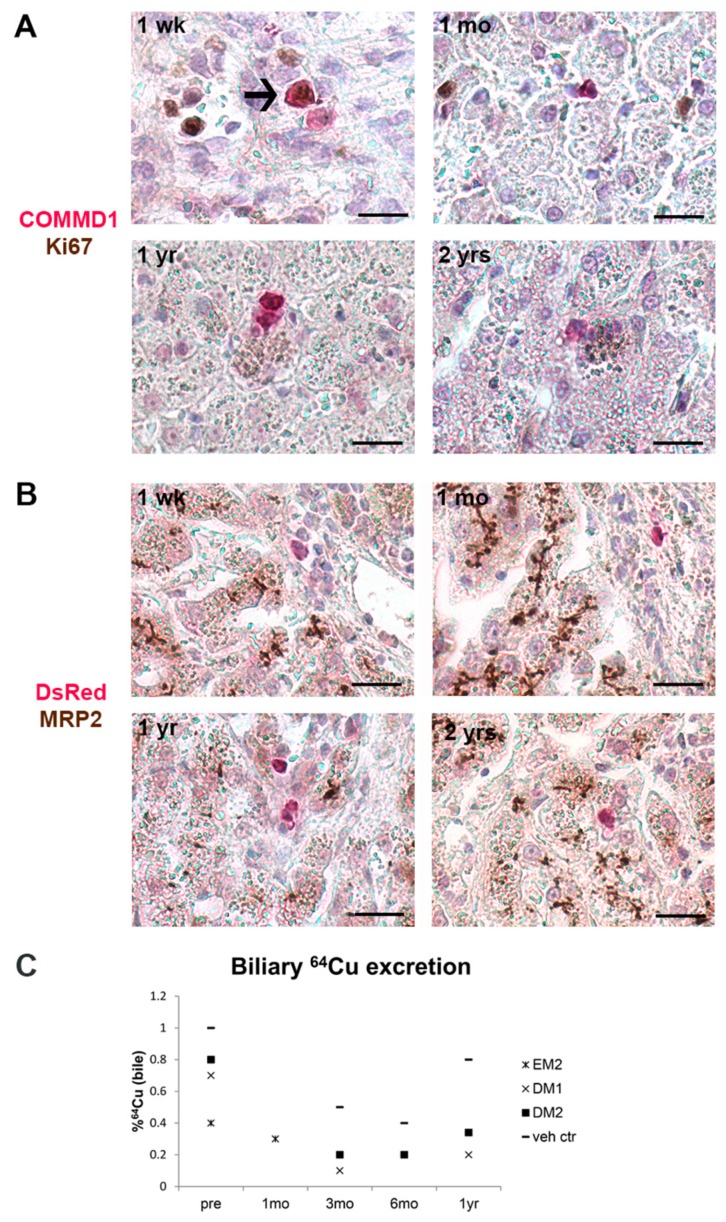
Proliferation and differentiation of transplanted cells in vivo. Double immunohistochemical staining was performed on liver sections post-transplantation to investigate the presence of proliferation marker Ki67 and differentiation marker MRP2 on transplanted cells. (**A**) Transplanted cells were sporadically positive for Ki67 (arrow), but only in sections one week after transplantation and not at later time points. (**B**) Transplanted cells did not show immunostaining for MRP2, whereas hepatocytes showed positive canalicular staining. 1 wk: one week (dog EM1); 1 mo: one month (dog EM2); 1 yr: one year (dog DM1); 2 yrs: two years (dog DM2) post- transplantation. Scale bars represent 20 µm. (**C**) Biliary copper excretion before (pre) and after transplantation indicates that the copper excretion remains low with all tested conditions. Pre: before transplantation; 1 mo: one month; 3 mo: three months; 6 mo: six months; 1 yr: one year after transplantation.

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
