# Peer review of "Long-Term Survival of Transplanted Autologous Canine Liver Organoids in a COMMD1-Deficient Dog Model of Metabolic Liver Disease"

_cells, 2020, doi:10.3390/cells9020410_

Round 1

Reviewer 1 Report

Authors tried to show long term engraftment of genetically corrected canine liver organoid in COMMD1 deficient dog model of metabolic liver disease.  

Even though engraftment rate is very low(0.6%-1.2%), long term survival of canine model was found and it was very interesting.

However there is no data about survival analysis of transplanted canine models compared to non transplanted models. Author should show survival analysis which can show clear advantage of organoid transplantation for large animals. In terms of in vitro experiment, organoid culture method and lentiviral transduction should be more mentioned. (organoid culture period, MOI etc) evidence of  engraftment should be shown more objective methods. immunostaining is not enough for us to believe actual engraftment of liver organoid.  

Reviewer 2 Report

The submission by Kruitwagen et. al., examines the in vivo survival of autologous, organoid-derived, hepatocyte-like cells that were transduced and selected for COMMD1 expression. The study asks whether 1) The isolated cells from autologous organoid culture can be transduced to express COMMD1, selected and expanded quickly enough to transplant enough cells; 2) What would be the successful method of transplantation; 3) whether repeat delivery could increase the percentage of transplanted cells; 4) whether the cells would repopulate and integrate into the liver; 5) Make any functional difference to subvert copper accumulation. They showed that the cells could be transplanted via the portal vein repeatedly and whether engrafted cells can survive long term in the canine liver. Since the canine model is good (apart from the non-human primates) for hepatocyte transplantation studies and efforts to circumvent the issue of immune suppression, the work should be of broad interest.

Critique:

1) In the main manuscript, authors should add a panel of western blots or other alternative experiments to validate the status of the undifferentiated and differentiated cells before and after the 12- week cell expansion.

2) The authors should add the functional state of the cells after the 6-week expansion.

3) Although the technique was elegant and there were detectable transplanted cells, they neither proliferated nor differentiated after engraftment long term as documented in figure 5. The co-staining for COMMD1 and Ki67 was undetectable. Thus there was no functional improvement on copper excretion. That is a concern.

4) The number of n in this study is small, particularly for the 2-year time point.

Round 2

Reviewer 1 Report

Authors have corrected and added the manuscript by recommendation by reviewer.

Author Response

We thank the reviewer for the valuable suggestions that improved the manuscript. 

Reviewer 2 Report

They have responded to a majority of the concerns.

I have a minor point: The authors have included data in supplementary figure 1 showing increased expression of HNF4a and albumin (ALB) in two donors indicated a successful hepatic differentiation. But this was only after 9-days under DM conditions as indicated in the figure legend. In the response letter, however, they mentioned that these gene expression analysis graphs were from the cells before transplantation. Please clarify the concern.

Author Response

We thank the reviewer for the overall comments and suggestions. We believe they improved the manuscript and overall readability. With regards to the final remark of the reviewer we can confirm that the differentiation was measured before transplantation in the two donors (DM) that were used. To make this clear in the manuscript we have adapted the following sentence: 'Differentiation of DM (DM, n=2 dogs) conditions before transplantation was confirmed by gene expression profiling indicating a decrease of stemness marker (LGR5) and an increase of hepatic markers (HNF4A and ALB) after differentiation, see Figure S1.'